

# Local short-term variability in solar irradiance

Gerald M. Lohmann[1], Adam H. Monahan[2], and Detlev Heinemann[1]

[1]Energy Meteorology Group, Institute of Physics, Oldenburg University, Germany
[2]School of Earth and Ocean Sciences, University of Victoria, Canada

*Correspondence to:* Gerald M. Lohmann (gerald.lohmann@uni-oldenburg.de)

**Abstract.** Characterizing spatio-temporal irradiance variability is important for the successful grid integration of increasing numbers of photovoltaic (PV) power systems. Using 1 Hz data recorded by as many as 99 pyranometers during the HD(CP)² Observational Prototype Experiment HOPE, we analyze field variability of clearsky index $k^*$ (i.e. irradiance normalized to clearsky conditions) and sub-minute $k^*$ increments (i.e. changes over specified intervals of time) for distances between tens of meters and about ten kilometers. By means of a simple classification scheme based on $k^*$ statistics, we identify overcast, clear and mixed sky conditions, and demonstrate that the last of these is the most potentially problematic in terms of short-term PV power fluctuations. Under mixed conditions, the probability of relatively strong $k^*$ increments of $\pm 0.5$ is approximately twice as high compared to increment statistics computed without conditioning by sky type. As well, spatial autocorrelation structures of $k^*$ increment fields differ considerably between sky types. While the profiles for overcast and clear skies mostly resemble the predictions of a simple model published by Hoff and Perez (2012), this is not the case for mixed conditions. As a proxy for the smoothing effects of distributed PV, we finally show that spatial averaging mitigates variability in $k^*$ less effectively than variability in $k^*$ increments, for a spatial sensor density of $2 \ \mathrm{km}^{-2}$.

## 1 Introduction

The number of photovoltaic (PV) power systems has drastically increased in many regions of the world during the last decade, reaching a total nominal capacity of more than 178 GW installed world wide at the end of 2014. The future global PV capacity is expected to continually increase further, with predictions for 2019 ranging from 396 through 540 GW (Solar Power Europe (SPE), 2015). In consequence, the challenges associated with the inherent volatility of PV power production and its fundamental cause, weather-induced heterogeneity in solar irradiance fields, will considerably increase as well (Stetz et al., 2015). Variability in both irradiance and irradiance increments (changes over specified intervals of time) are of interest in this context. On the one hand, variability in irradiance itself primarily affects the yield of a PV system and the dimensioning of battery storage. On the other hand, variability in irradiance increments impacts the balancing of generation and load, as well as the maintenance of power quality such as voltage and frequency stability. Depending on the dimensions of the power grid and the PV capacity in question, relevant variability in irradiance and its increments can span a broad range of spatio-temporal scales, from seconds and meters up to days and hundreds of kilometers. There is a need to understand the biases in representation of temporal variability resulting from temporally coarse-resolution observations (Yordanov et al., 2013b), as well as how spatial averaging (as would come from having distributed PV over a region) mitigates variability (Hoff and Perez,





2010). Characterizing the spatio-temporal volatility of irradiance fields and their increments is key to the planning and reliable operation of future power grids and their corresponding subsystems.

Recent studies of PV-related variability have analyzed power spectra of PV systems and solar irradiance (Calif et al., 2013; Curtright and Apt, 2008; Klima and Apt, 2015; Lave and Kleissl, 2010; Marcos et al., 2011a; Tabar et al., 2014; Yordanov et al., 2013b), compared power fluctuations from specific PV plants with corresponding irradiance measurements (Lave and Kleissl, 2013; Lave et al., 2013; Marcos et al., 2011a; van Haaren et al., 2014), and characterized power variability as a function of PV plant size (Dyreson et al., 2014; Lave et al., 2012; Marcos et al., 2011b; van Haaren et al., 2014). Furthermore, spatial autocorrelation structures and decorrelation length scales of increments in irradiance and clearsky index (i.e. irradiance normalized to clearsky conditions), and PV power output have also been studied for a range of spatial scales and increment values (Arias-Castro et al., 2014; Elsinga and van Sark, 2014; Hinkelman, 2013; Hoff and Perez, 2012; Lave and Kleissl, 2013; Mills, 2010; Perez et al., 2012; Perpiñán et al., 2013). For all quantities and methods considered, increment correlations at different locations have been shown to decrease with increasing distance, with a smaller rate of decrease (longer decorrelation distances) for larger time increments.

While satellite-derived irradiance data are convenient for the analysis of large spatio-temporal scales, comprehensive datasets for local short-term variability are time-consuming and expensive to collect, and although needed have not previously been available (Hoff and Perez, 2012; Perez et al., 2012). Therefore, previous studies are either restricted to a limited spatial resolution, a limited temporal resolution, or both. For example, the few studies that have been based on high-resolution 1 Hz PV power observations only had measurements from a maximum of six PV systems (Marcos et al., 2011a, b) at their disposal. Though irradiance measurements with this high temporal resolution have on occasion been conducted with more sensors, the spatial coverage remains strongly confined (e.g. up to 45 sensors spread across $\sim 2.5$ km$^2$; Dyreson et al., 2014). Some studies have used artificially generated data to overcome these restrictions, either by simulating simple cloud shapes (Arias-Castro et al., 2014; Lave and Kleissl, 2013), or by constructing virtual networks based on time shifted single sensor measurements (Perez et al., 2012). However, these simulated datasets do not necessarily coincide with reality.

To fill the gap in understanding of small-scale spatial and temporal variability in irradiance, we use an extensive experimental dataset of global horizontal irradiance (GHI) field samples from two measurement campaigns to characterize sub-minute variability of clearsky index for distances between tens of meters and about ten kilometers. A high temporal resolution of 1 Hz and the use of up to 99 synchronized silicon photodiode pyranometers yields a robust basis for the analyses (these data are described in detail in section 2). Based on this dataset with its unprecedentedly fine resolution in both space and time, we study single-point statistics and two-point correlation coefficients of clearsky index, and develop a simple classification scheme to identify overcast, clear and mixed skies (section 3). Conditioned on these three sky types, we then analyze the probability distributions of sub-minute increments in clearsky index for single sensors and large spatial averages of about 80 km$^2$, as well as their corresponding spatial autocorrelation structures (section 4). Finally, we spatially average randomly selected sensors from the dataset, covering different area sizes but maintaining a fixed spatial density, as a proxy for the smoothing effects of distributed PV power production (section 5). Discussions and conclusions follow in sections 6 and 7.



## 2 Data

### 2.1 Measurement campaigns

The datasets on which this study's analyses are based originate from two extensive measurement campaigns performed during the HD(CP)$^2$ Observational Prototype Experiment (HOPE) using a set of autonomous silicon photodiode pyranometers. These instruments measure the downwelling shortwave radiation at the Earth's surface in the range between 0.3 and 1.1 μm. Although this wavelength band does not span the entire solar irradiance spectrum, it corresponds well with the relevant bandwidths of typical semiconductor materials used for photovoltaic applications (Pérez-López et al., 2007). In fact, the pyranometers themselves may essentially be thought of as tiny PV systems, reduced in space to a single point. Equipped with a battery power supply for up to 10 days, they store their data on-site with a temporal resolution of 10 Hz (averaged to 1 Hz during post-processing). A GPS-based clocked control is used to ensure synchronization between sensors, and for proper positioning data.

The first field campaign with these instruments took place near Jülich, Germany (50.9° N, 6.4° E), from April 2 through July 24, 2013. It featured a total of 99 pyranometers deployed over an area of about 80 km$^2$. The second was conducted near Melpitz, Germany (51.5° N, 12.9° E), and lasted from September 3 until October 14, 2013. During this time, 50 pyranometers, all of which had previously been used in the Jülich campaign, were deployed over an area of about 4 km$^2$. During the measurement campaigns each instrument was subject to regular weekly maintenance. This maintenance included data transfer, battery replacement, thorough cleaning of the glass dome, and re-leveling of the mounting platform (if necessary). As part of this process, the states of cleanliness and orientation were recorded, in order to facilitate identification of periods of bad data. For each observation of tilt or fouling, that week's worth of data was flagged accordingly, even though the specific problem did not necessarily last for the entire preceding week.

The geometry of the pyranometer locations for the Melpitz and Jülich campaigns, as well as a histogram of all sensor pair distances $d_{ij}$ is presented in Fig. 1. The sensor layout of the Melpitz campaign, with many sensors concentrated in the center and fewer towards the edge of the domain, is more structured than that of the Jülich campaign. This difference is due to the much larger spatial domain in the Jülich case, which entailed external restrictions on the instrument locations, such as road access, setup permission, and agricultural land use. Consequently most of the very short sensor pair distances ($d_{ij} < 1$ km), and some intermediate distances (1 km $< d_{ij} < 2$ km) are attributed to the Melpitz campaign, while sensor pair counts with $d_{ij} > 3$ km are all associated with the Jülich campaign (Fig. 1c).

Taking into account the final datasets' high temporal resolution of 1 Hz, along with the corresponding dense spatial coverages, the two field campaigns provide the basis for unique analyses of irradiance variability, particularly regarding potential PV power fluctuations. Schmidt et al. (2015) use data from the Jülich campaign for a performance evaluation of sky imager based solar irradiance forecasts, and Madhavan et al. (2015) present a more detailed discussion of the campaign and the instrumentation. To the best knowledge of the authors, no other PV-related studies based on comparably dense and high-frequency irradiance sensor networks have been published to date.





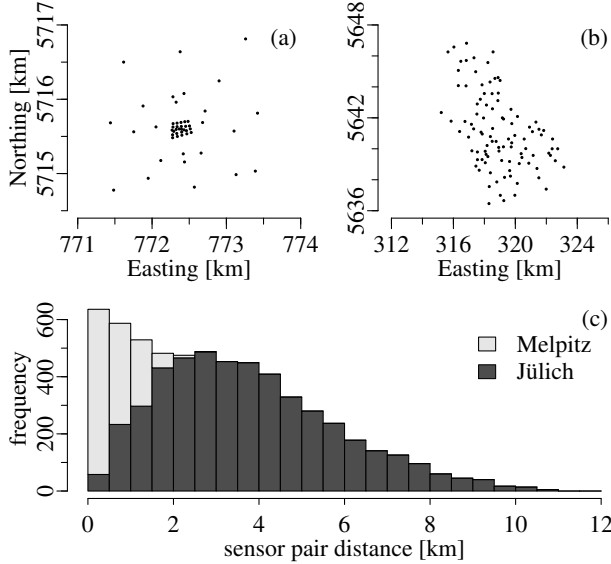

**Figure 1.** Panels (a) and (b) show maps of the coordinates (UTM projection, grid zone 32U) of all pyranometers deployed during the two field campaigns in (a) Melpitz and (b) Jülich. Panel (c) displays a histogram of the combined station pair distances from both data sets.

## 2.2 Clearsky index

The available global horizontal irradiance (GHI) at any given point on the Earth's surface is subject to influences from both astronomical and atmospheric processes. As for the former, the apparent movement of the sun relative to Earth gives rise to diurnal and seasonal variations in GHI. These variations are accurately predictable and not large on short time scales of

seconds or minutes. On the other hand, weather-related contributions to irradiance variability are manifold and complex, and present on all time scales. For instance, the growth, motion, and decay of clouds can affect the seasonal cycle in GHI (e.g. winter tends to be cloudier than summer in mid-latitude low-lying land), and the rapid succession of sunlight exposure and cloud shadow in conditions dominated by fair-weather cumulus generates stochastic variability on short time scales (seconds – minutes) (Woyte et al., 2007). The presence of different kinds of cloud (e.g. layer vs. convective) at different altitudes and

of different composition (e.g. high-albedo small cloud droplets vs. low-albedo large droplets in rainclouds) result in a complex set of influences on GHI over a broad range of timescales.

In order to distinguish the cloud-induced fluctuations from the slowly-evolving, astronomically-determined apparent motion of the sun, a GHI time series $G$ at a particular location may be related to either the extraterrestrial solar radiation $G_{extra}$ (i.e. irradiance on Earth if there were no atmosphere), or the clearsky radiation $G_{clear}$ (i.e. irradiance on Earth with a cloud-

free atmosphere). Knowing these, we can define the clearness index

$$k = \frac{G}{G_{extra}}, \tag{1}$$





and clearsky index

$$k^* = \frac{G}{G_{clear}}, \tag{2}$$

respectively. The extraterrestrial solar radiation $G_{extra}$ depends only on astronomical relationships, whereas a calculation of clearsky radiation $G_{clear}$ requires parameters of atmospheric conditions, such as typical water vapor concentration and aerosol
load.

While all atmospheric influences on GHI variability are included in $k$, variations in the clearsky index are dominated by changes in cloud cover. Other sub-daily variations, especially those caused by changes in light scattering with solar angle $\alpha$, are ideally removed entirely from $k^*$ (although depending on how accurately the atmospheric conditions and their variability are actually estimated, such changes with $\alpha$ may still affect the clearsky index to a minor degree). With $k^*$ thus being better
suited to remove trends in GHI variability, the focus of this analysis will be clearsky irradiance time series computed for the respective locations of both field campaigns, using the clearsky model described by Fontoynont et al. (1998). A limitation of this model is that it is based on climatological means and does not account for all variations in scattering or absorption properties. Moreover, relatively low values of GHI occurring shortly after sunrise and just before sunset, coupled with path prolongation and corresponding higher uncertainties in clearsky irradiance calculations at these times, can result in unrealistic
values of $k^*$ (Lave et al., 2012). In consequence, we only consider data associated with $\alpha > 15°$ throughout the study. While the resulting clearsky index remains an approximate model-based quantity, rather than a direct measurement, it allows us to focus directly on weather-related variations in surface irradiance.

The lowest values of $k^*$ are typically not zero, because even the darkest of clouds do not attenuate all irradiance. As well, the upper limit can exceed one, primarily due to short-term reflections from the sides of clouds (and also to a secondary degree due
to the limitations of the clear sky model). Under broken cloud conditions this phenomenon, known as cloud enhancement, can cause single-point GHI to exceed its corresponding clearsky irradiance value by more than 50 % on short time scales (Luoma et al., 2012; Piacentini et al., 2011; Yordanov et al., 2013a).

To characterize the modulation of $k^*$ variability by the prevailing sky type (e.g. overcast vs. clear sky), we divide the time series at each sensor into non-overlapping 15-minute windows. This sub-hourly timescale is short enough that it is typically
dominated by a single sky type, but long enough that there is enough variability to make statistical analyses meaningful. We will use differences in the statistics of $k^*$ within these 15-minute windows to define different sky type categories.

To illustrate the wide range of cloud influences on $k^*$ statistics in these 15-minute windows, Fig. 2 presents three distinct examples of spatio-temporal variability in $k^*$. These representative subsets have been manually selected from a pool of random windows sampled from the entire duration of the Jülich campaign. Each panel includes summary statistics for all sensors in the
domain for the period (represented as box plots), as well as the variability of a single randomly-selected sensor. The boxplots each consist of a lower "whisker" $w_{low}$, the first quartile $Q_1$, the median $Q_2$, the third quartile $Q_3$, and an upper "whisker" $w_{up}$ (summarized in Table 1). Following common practice when presenting box plots, $w_{low}$ ($w_{up}$) is defined as the lowest (highest) data point that still falls within the range of $Q_1 - 1.5 \cdot IQR$ and $Q_3 + 1.5 \cdot IQR$, with $IQR$ denoting the interquartile range $IQR = Q_3 - Q_1$ (Devore, 2015). Any data below $w_{low}$ or above $w_{up}$ are considered outliers.




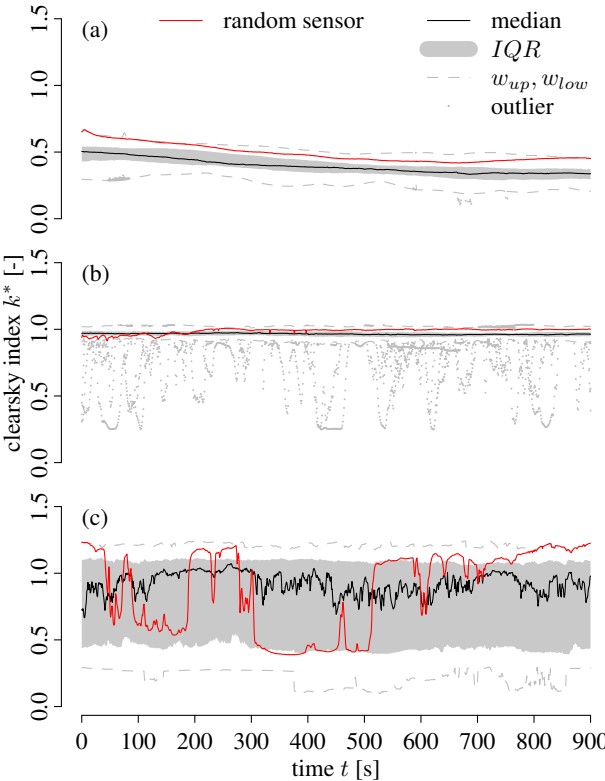

**Figure 2.** Examples of different spatio-temporal variability in clearsky index $k^*$ for three distinct cases of sky types: (a) mostly overcast, (b) mostly clear, and (c) mixed. These representative subsets have been manually selected from the Jülich campaign and span 15 minutes each. The time series of randomly selected sensors (red curves) are contrasted with summary statistics of field variability, represented as box plots (Table 1).

**Table 1.** Summary statistics used to visualize data spread throughout this study.

| Name | Symbol | Definition |
|---|---|---|
| First quartile | $Q_1$ | 75 % of the data $> Q_1$, and 25 % $< Q_1$ |
| Median (second quartile) | $Q_2$ | 50 % of the data $> Q_2$, and 50 % $< Q_2$ |
| Third quartile | $Q_3$ | 25 % of the data $> Q_3$, and 75 % $< Q_3$ |
| Interquartile range | $IQR$ | $Q_3 - Q_1$ (interval containing half of the data) |
| Lower whisker | $w_{low}$ | lowest value $> Q_1 - 1.5 \cdot IQR$ |
| Upper whisker | $w_{up}$ | highest value $< Q_3 + 1.5 \cdot IQR$ |
| Outliers | - | any data points $< w_{low}$ or $> w_{up}$ |





The 15-minute window in panel 2(a) features very little spatial variability and a continually low range of $k^*$ values, corresponding to a time of overcast conditions during which a fairly homogeneous cloud layer spanned the entire domain. In contrast, the majority of sensors in panel 2(b) show a continually high range of $k^*$ with little spatial variability, with the exception of some pronounced rapid and short-lived decreases in $k^*$. Clear sky conditions dominated the domain at this time, with occasional short-duration shadows cast on single sensors (although not the single example sensor). Finally, the data shown in panel 2(c) display considerable variability throughout the domain at all times, with a consistent $IQR$ value of $\sim 0.5$. The trace of the example sensor clearly illustrates the predominant condition of mixed skies in this case, with an alternation between cloud coverage and clear sky exposure. The characteristic differences in the temporal average and variability of $k^*$ evident in these example datasets indicate that a natural classification scheme for sky type within 15 minutes can be developed in terms of the statistics of $k^*$.

## 3 Sky type variability

### 3.1 Single-point statistics of clearsky index

In order to assess the character of irradiance variability conditioned on sky type, we group subsets of similar sky conditions by means of two simple statistics. Specifically, we compute the sample arithmetic mean

$$\overline{k_i^*} = \frac{1}{N} \sum_{t=1}^{N} k_i^*(t), \tag{3}$$

and the sample standard deviation

$$\sigma_i^{k^*} = \sqrt{\frac{1}{N-1} \sum_{t=1}^{N} (k_i^*(t) - \overline{k_i^*})^2} \tag{4}$$

of $k_i^*$ (the clearsky index of the $i^{\text{th}}$ sensor) for each sensor for all 15-minute periods, using non-overlapping windows of width $T = 900\,\text{s}$ (resulting in a sample number $N = 900$ for the 1 Hz data).

These two statistical measures allow an intuitive characterization of the prevailing sky type that a sensor has been subjected to for a limited time, by quantifying the average and spread of the respective 15-minute window in its timeseries. A kernel density estimate (KDE) of the joint probability density function (PDF) of $\overline{k_i^*}$ and $\sigma_i^{k^*}$ using data for all individual sensors and all available days from both measurement campaigns is presented in Fig. 3. The estimated PDF is overlaid with a regular grid that we will use to define particular sky type classes.

In the low variability range $\sigma_i^{k^*} < 0.09$, the two peaks in the joint PDF (located in A1/B1, and D1/E1 of Fig. 3, respectively) clearly represent the cases of predominant overcast (low $\overline{k_i^*}$ and low $\sigma_i^{k^*}$), and clear sky (high $\overline{k_i^*}$ and low $\sigma_i^{k^*}$) conditions. In addition to these two well-defined end members, intermediate sky conditions were also recorded by the single sensors. While these span the entire ranges of $\overline{k_i^*}$ and $\sigma_i^{k^*}$, they are not uniformly distributed. In fact, a separation between relatively clear sky and overcast conditions (i.e. high and low $\overline{k_i^*}$) can be observed for most values of $\sigma_i^{k^*}$, though the distinction becomes less pronounced with increasing $\sigma_i^{k^*}$.





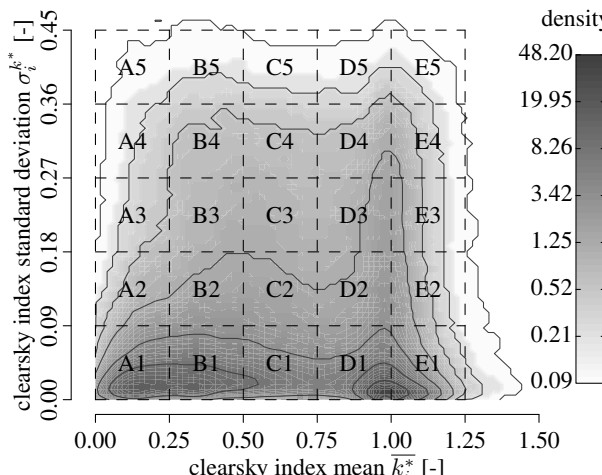

**Figure 3.** Kernel density estimate (KDE) of the joint probability density function (PDF) of mean $\overline{k_i^*}$ and standard deviation $\sigma_i^{k^*}$ of clearsky index, based on all available single sensor data from both measurement campaigns. The overlaid regular grid is used to define particular sky type classes.

## 3.2 Two-point correlations of clearsky index

The quantities $\overline{k_i^*}$ and $\sigma_i^{k^*}$ provide single-point temporal statistical information about variability in $k^*$. As a first characterization of the spatial structure of the $k^*$ field, we compute the spatial two-point correlation coefficients between sensors i and j

$$\rho_{ij}^{k^*} = \frac{\sum_{t=1}^{N}(k_i^*(t) - \overline{k_i^*})(k_j^*(t) - \overline{k_j^*})}{\sqrt{\sum_{t=1}^{N}(k_i^*(t) - \overline{k_i^*})^2 \sum_{t=1}^{N}(k_j^*(t) - \overline{k_j^*})^2}} \tag{5}$$

conditioned on the 15-minute intervals falling within individual boxes in Fig. 3. That is, all 15-minute time windows, during which both sensors in a pair are simultaneously associated with the same sky type, are used to calculate a single $\rho_{ij}^{k^*}$ value for the respective pair of sensors. In Eq. (5), $k_i^*(t)$ and $k_j^*(t)$ are the two individual time series of the pair of sensors for those time periods in which the statistics of both sensors are in the same grid box, while $\overline{k_i^*}$ and $\overline{k_j^*}$ are the corresponding arithmetic means. The quantity $N$ denotes the number of data points in the two time series. Note that the meaning of the overbar, denoting the arithmetic mean, is different than that in Eq. (3) and 4, as the averaging time increases from 900 s in Eq. (3) to the total time of simultaneously available 1 Hz data of $i$ and $j$ in Eq. (5).

The resulting distributions of spatial autocorrelation functions $\rho_{ij}^{k^*}$ are shown as functions of sensor pair distance $d_{ij}$ in Fig. 4 for each of the grid boxes from Fig. 3. The same box plot statistics listed in Table 1 are computed for 10 logarithmically





spaced bins of $d_{ij}$. A pair of sensors is only included in these calculations if its members share the same sky type for at least 60 minutes over the observational period. The number of pairs used to derive the box plot information is given in each panel.

Pairs of sensors with very high $\sigma_i^{k^*}$ and very low $\overline{k_i^*}$ (panels A3 through A5), are found to be virtually non-existent, although these ranges are occupied by individual sensors (cf. Fig. 3). Similarly, combinations with relatively high $\sigma_i^{k^*}$ and moderate or large $\overline{k_i^*}$ (panels B4/5 and E4/5) also lack a high number of available sensor pairs. The remaining well sampled grid boxes all show spatial autocorrelation functions $\rho_{ij}^{k^*}$ that decrease with increasing $d_{ij}$, as expected. However, the rates of decrease vary considerably across the different grid boxes. The differences in autocorrelation structure between two adjacent grid boxes (e.g. A1 and B1) are generally small, but become more pronounced when comparing those farther apart (e.g. A1 and D4).

For further analyses, and consistent with the manually selected exemplary periods previously shown in Fig. 2, we consider a classification of three distinct sky types based on the grid boxes:

1. overcast (A1 and B1),

2. clear (D1 and E1), and

3. mixed (A3 through E5).

This classification is based upon the subjective identification of different grid boxes in Fig. 3 and 4 with characteristic statistical properties. Although some data corresponding to intermediate sky conditions (panels A2 through E2, as well as C1) are neglected using this classification scheme, the structure of $\rho_{ij}^{k^*}$ is appreciably distinct for all three identified sky types.

Under overcast conditions, correlation coefficients remain $\rho_{ij}^{k^*} \approx 1$ for $d_{ij} \lesssim 1$ km, while clear conditions deviate from $\rho_{ij}^{k^*} \approx 1$ even for relatively small separations $d_{ij} \lesssim 0.05$ km. Correlation values $\rho_{ij}^{k^*} \approx 1$ appear under mixed sky conditions only for very small $d_{ij} \lesssim 0.05$ km. With increasing distances, the three sky types' spatial autocorrelation structures also differ in their rates of decay. For example, the characteristic distance to reach $\rho_{ij}^{k^*} \approx 0.5$ is about 10 km for clear sky and overcast conditions, while it is an order of magnitude smaller ($\sim 1$ km) for the mixed sky type.

The $k^*$ autocorrelation structures within the different sky types are consistent with the associated cloud patterns. The results for overcast and clear sky conditions both suggest fairly large and homogeneous structures (cf. 10 km to reach $\rho_{ij}^{k^*} \approx 0.5$), corresponding to large stratus-type cloud layers in the former case, and homogeneously clear skies (with infrequent and localized shadowing of the sensors) in the latter. During times classified as mixed, the structure of $\rho_{ij}^{k^*}$ indicates that heterogeneous cloud fields dominate, with much smaller length scales than those under overcast conditions. The decay length scale of correlations under mixed skies corresponds well with typical cloud length scales $\lesssim 2$ km of cumulus-type clouds (Neggers et al., 2003).

## 4 Variability in clearsky index increments

The previously discussed properties of the observed $k^*$ fields are independent of their ordering in time, i.e. randomly shuffling all sensor-pair data in time (within the 15-minute windows) will result in the same spatial autocorrelation structures. While this overall variability is of some interest (e.g. when considering long-term yield of PV systems), it does not characterize how





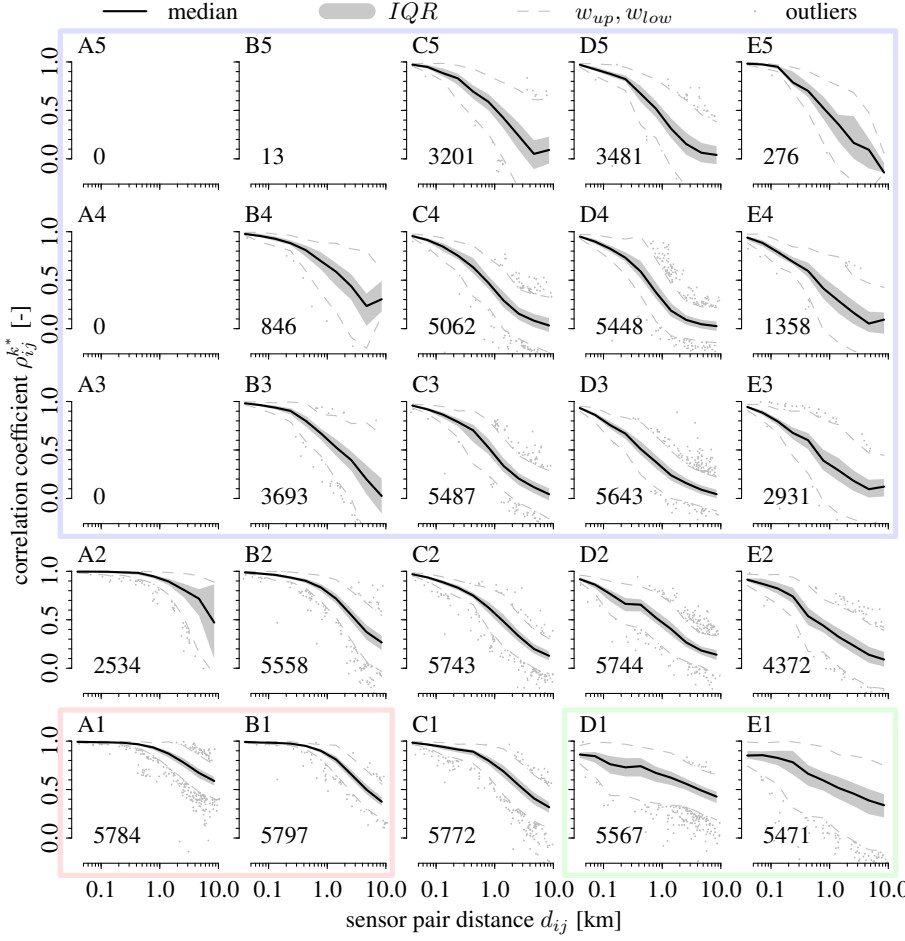

**Figure 4.** Spatial two-point correlation coefficients $\rho_{ij}^{k^*}$ of clearsky index $k^*$ as functions of sensor pair distance $d_{ij}$ for each of the grid boxes from Fig. 3, based on data from both field campaigns. The summary statistics (cf. Table 1) are based on 10 logarithmically scaled bins of $d_{ij}$, and include only those sensor pairs whose members simultaneously correspond to the same grid box for at least 60 minutes. The total number of pairs used to derive the statistics is given in each panel. Grid boxes to be subsequently grouped as similar sky types are indicated by colored boxes.

rapidly $k^*$ fields can change. A useful measure of intermittency in the clearsky index is the statistics of $k^*$ increments

$$\Delta k_\tau^*(t) = k^*(t + \tau) - k^*(t) \tag{6}$$

for different time lags $\tau$.



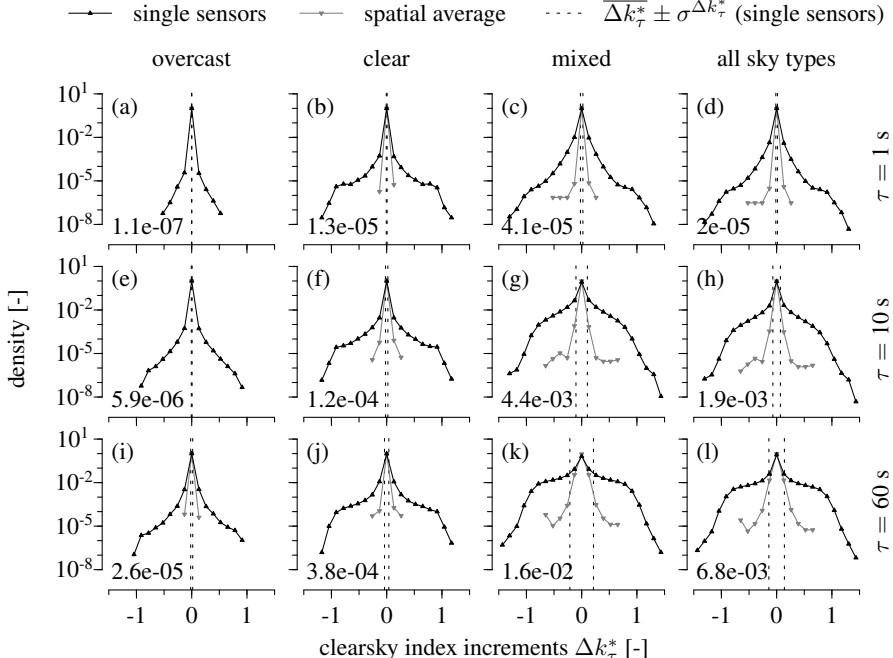

**Figure 5.** Statistics of clearsky index increments $\Delta k_\tau^*$ for different time lags $\tau$, based on data from the Jülich campaign. The first three columns display distributions conditioned on different sky types, while the rightmost column presents the combined statistics for all sky types. The estimated probability density functions of all single sensors (solid black lines) are supplemented with the range of $\pm$ one standard deviation $\sigma^{\Delta k_\tau^*}$ around their means $\overline{\Delta k_\tau^*}$ (dashed black lines), and contrasted to the much narrower distributions of the spatially smoothed average of all pyranometers (solid gray lines). The respective average probabilities $P(\Delta k_\tau^* = \pm 0.5)$ of single sensors to fluctuate by $\pm 0.5$ are quoted in each panel.

## 4.1 Increment statistics

PDFs of $\Delta k_\tau^*$ estimated from data from the Jülich campaign for three short-term time lags $\tau = (1,\ 10,\ 60)$ s are presented in Fig. 5. The results are first conditioned on the sky type classification (first three columns) and then shown again for all sky types together (rightmost column). As a first illustration of the effect of spatial averaging on fluctuation statistics, the PDF of

5 area-averaged increments is also included in each panel.

All PDFs are characterized by a narrow central peak - corresponding to a high probability of very small increments - surrounded by broad tails in which the PDF decreases slowly. With increasing $\tau$, these tails become flatter and the probability of very large excursions increases. While the PDFs of single sensor observations all exhibit such tails, these features are much less prominent for the spatially-averaged $k^*$ increment distributions.

10 Under overcast conditions, the central peaks of the distributions are generally prominent and the tails are not particularly pronounced. The PDFs of clear skies also have a strong central peak but display broad flat tails (higher than for overcast





conditions), representing the rare large excursions evident in Fig. 2. The central peak is wider under mixed conditions, and the flanks are flatter than for clear skies. Under mixed conditions, probabilities of large excursions are enhanced by the rapid changes associated with passing cloud edges. This distribution of increments is consistent with the individual time series shown in Fig. 2. The statistics obtained when all sky types are taken together feature the same shapes in the tails as those of the mixed

sky type, because the extreme fluctuations in $k^*$ are most common under mixed conditions.

A measure of the extent of the tails of the PDF is the probability $P(\Delta k_\tau^* = \pm 0.5)$ of a single sensor to fluctuate by $\pm 0.5$. The values for this quantity, quoted in each panel of Fig. 5, take different orders of magnitude among the different sky types. These probabilities increase from overcast to clear and then to mixed sky conditions, while the values associated with the overall statistics of all sky types are located somewhere in between the last two classifications. Compared to the statistics of

all sky types, and independent of $\tau$, $P(\Delta k_\tau^* = \pm 0.5)$ is more than twice as high under conditions classified as mixed. Thus, for applications such as the maintenance of grid stability, where worst case scenarios (in terms of strong short-term PV fluctuations) are of interest, the conditioning of $\Delta k_\tau^*$ statistics on different sky types demonstrates the strong dependence on specific sky conditions of the likelihood of severe fluctuations occurring shortly one after another.

While changes in increment variability are reflected by changes of $\sigma^{\Delta k_\tau^*}$ (dashed lines in Fig. 5), the standard deviation is

not an appropriate measure of the size of extreme fluctuations due to the non-Gaussian character of $k^*$ increment statistics. In consequence, the widely used three-sigma rule of thumb, according to which a range of $\overline{\Delta k_\tau^*} \pm 3\,\sigma^{\Delta k_\tau^*}$ would cover 99.73 % of the values if $\Delta k_\tau^*$ were normally distributed (M.S. Nikulin (originator), 2002), can be misleading when applied to $k^*$ fluctuations. For example, only about 95 % of the empirical single sensor data is included in this range for the most variable subset of $\tau = 60$ s and mixed skies (Fig. 5k). This result is in line with previous findings of the 99.7[th] percentile of one minute

increments in clearsky index being about seven standard deviations away from the mean (Mills, 2010).

## 4.2 Two-point correlations of increments

The differences between the single sensor increment statistics and the distributions of areal averages in Fig. 5 are of interest, because the spatial averaging clearly results in a substantial reduction in the probability of high-magnitude $k^*$ fluctuations. In order to specify the underlying spatio-temporal field characteristics, we analyze two point correlation coefficients of $k^*$

increments $\rho_{ij}^{\Delta k_\tau^*}$ (computed as in Eq. 5), based on data from both field campaigns. Various models have previously been proposed to predict the behavior of increment correlation as a function of distance for specific temporal scales, either by using empirical fits to measured data (Hoff and Perez, 2012; Lave and Kleissl, 2013; Lonij et al., 2013; Mills, 2010; Perpiñán et al., 2013) or by modeling simplified cloud shapes (Arias-Castro et al., 2014). While the empirical fits are based on datasets of limited spatio-temporal resolutions, the more theoretical model does not account for the complexity encountered in real cloud

shapes. Our dataset permits a direct empirical assessment of the spatial correlation structure of increments.

Conditioned on the previously defined classification scheme of sky types, summary statistics of $\rho_{ij}^{\Delta k_\tau^*}$ (cf. Table 1) are presented as functions of sensor pair distance for different time lags in Fig. 6, using 10 logarithmically scaled $d_{ij}$ bins. Also shown in Fig. 6 are the autocorrelation structures as functions of sensor pair distance divided by increment size, obtained





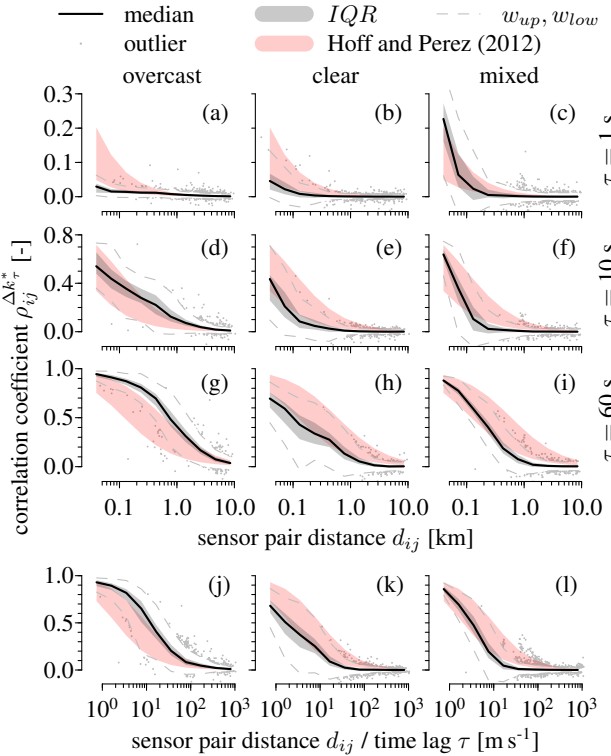

**Figure 6.** Spatial two-point correlation coefficients $\rho_{ij}^{\Delta k_\tau^*}$ of clearsky index increments $\Delta k_\tau^*$ as functions of sensor pair distance $d_{ij}$ for different time lags $\tau$ (a – i), as well as sensor pair distance divided by time lag (j – l). The results are conditioned on different sky types and based on data from both field campaigns. The summary statistics (cf. Table 1) are computed with 10 logarithmically scaled $d_{ij}$ bins, and include only those sensor pairs whose members simultaneously correspond to the same sky type for a total of at least 60 minutes. The colored regions correspond to the model $\rho_{ij}^{\Delta k_\tau^*} = (1 + \frac{d_{ij}}{\tau \cdot v})^{-1}$ (Hoff and Perez, 2012), using a range of relative cloud speeds $2\,\mathrm{ms}^{-1} < v < 10\,\mathrm{ms}^{-1}$. Note the different scales on the vertical axes.

using $\Delta k_\tau^*$ time series of all three increments $\tau = (1, 10, 60)$ s. The plots only include those sensor pairs whose members simultaneously correspond to the same sky type for a total of at least 60 minutes.

A useful measure of increment correlation structure is the decorrelation length scale $d_0$, which we define to be the minimum distance at which $\rho_{ij}^{\Delta k_\tau^*} = 0.05$. These distances generally increase with increasing time lags, and they decrease from overcast
5   to clear skies, and then to mixed conditions. As an exception to the latter statement, under overcast and clear conditions $k^*$ increments associated with very short time lags $\tau = 1$ s (panels a and b) are uncorrelated even for very small $d_{ij}$. The mixed sky type features a measurable decorrelation length scale $d_0 \approx 90$ m for $\tau = 1$ s. For $\tau = 10\,\mathrm{s}$ ($\tau = 60\,\mathrm{s}$), this distance increases to 2.2 km (7.2 km) under overcast, 0.2 km (1.5 km) under clear sky, and 0.2 km (1.1 km) under mixed sky conditions. These decorrelation distances estimated from Fig. 6 are all summarized in Table 2.





**Table 2.** Increment decorrelation length scales $d_0$ and effective decorrelation length scales $d_0' = \hat{d}_0 \cdot \tau$ (for which $\rho_{ij}^{\Delta k_\tau^*} = 0.05$) estimated from Fig. 6. Scaled decorrelation distances $\hat{d}_0$ used to derive $d_0'$ are obtained from the generalized analysis in said figure's panels (j) – (l).

| | overcast | | clear | | mixed | | $d = \tau \cdot v(\frac{1}{\rho} - 1)$ | | |
|---|---|---|---|---|---|---|---|---|---|
| $\tau$ [s] | $d_0$ | $d_0'$ | $d_0$ | $d_0'$ | $d_0$ | $d_0'$ | $d_0^{v=2\,\mathrm{ms}^{-1}}$ | $d_0^{v=10\,\mathrm{ms}^{-1}}$ | [km] |
| 1 | < 0.05 | 0.161 | < 0.05 | 0.028 | 0.092 | 0.016 | 0.038 | 0.190 | |
| 10 | 2.171 | 1.611 | 0.231 | 0.285 | 0.201 | 0.163 | 0.380 | 1.900 | |
| 60 | 7.169 | 9.664 | 1.541 | 1.708 | 1.077 | 0.977 | 2.280 | 11.400 | |

In addition to the statistics of the measured data, each panel of Fig. 6 also includes colored regions corresponding to the model of spatial correlation coefficients

$$\rho_{ij}^{\Delta k_\tau^*} = (1 + \frac{d_{ij}}{\tau \cdot v})^{-1} \tag{7}$$

proposed by Hoff and Perez (2012), using a range of cloud speeds $2\ \mathrm{ms}^{-1} < v < 10\ \mathrm{ms}^{-1}$. These speed values corre-
spond approximately to the range of mean vector winds at 850 hPa $v_{850}$ for Germany from the ERA-40 re-analysis atlas
($2\ \mathrm{ms}^{-1} \lesssim v_{850} \lesssim 8\ \mathrm{ms}^{-1}$; Uppala et al., 2005; ECMWF). While Hoff and Perez (2012) obtained Eq. (7) as a curve fit from
data with hourly resolution and pair distances up to several hundred kilometers, they also considered an initial assessment of
its applicability to higher spatio-temporal resolutions. Compared to other models, Lave and Kleissl (2013) showed Eq. (7) to
yield relatively high correlation coefficients and relatively long decorrelation length scales. Of the models considered by Lave
and Kleissl, the selected one provides the best fit to our data. As the model makes no distinction between sky types, its output
in Fig. 6 is the same for each row of panels and the quantity $v$ cannot be interpreted as the actual speed of the clouds. Rather,
it represents a quantity with the units of speed that combines information about cloud type and motion.

The ranges of the model results are in broad agreement with the summary statistics of the two field campaigns and the
general decrease of spatial correlation with increasing distance is reproduced well. However, differences between overcast and
clear sky conditions are evident, as the former tends to coincide with the upper region of the model range (corresponding to
high $v$), while the latter rather agrees with the lower end of the range (low $v$). Overcast conditions for $\tau = 1$ s again are an
exception to this general rule. For the mixed sky type, and for all values of $\tau$, the correlation values coincide with the upper
region of the model for very small $d_{ij}$, but they decrease more rapidly with increasing distances and thus feature decorrelation
distances that are consistently much smaller than the modeled ones.

Finally, the bottom panels (j) – (l) bring the above results together by presenting the correlation coefficients of different
sky types as functions of distance over time lag. Again, overcast conditions coincide with model outputs for high values of $v$
almost entirely, while results for clear skies mostly overlap with modeled correlation coefficients for low $v$. Mixed conditions
are highly correlated for short scaled distances (corresponding to the upper region of the model), but de-correlate rapidly with
increasing $d_{ij} \cdot \tau^{-1}$. If the range of $v$ values considered was broadened, the correlation structures would fall within the model
envelope for all sky types. However, while the profiles for overcast and clear skies look similar to the model prediction for





some specific $v$, this is not the case for the mixed cloud case. For short scaled distances the correlation decay corresponds to an intermediate value of $v$ in the model - but after that, the correlations drop off much faster. This result demonstrates that the model is not able to capture the correlation structure for mixed sky conditions.

## 5 Variability in spatial averages

Averaging clearsky index increments from different sensors provides an estimate of the output variability of an ensemble of PV installations at multiple locations. In order to assess the effect of area averaging on variability as a function of averaging area $A$, we employ the following random circle sampling method. First, the borders of the domain corresponding to each field campaign were determined by a convex hull encircling the instruments' coordinates, using the mean minimum distance between sensors as a circumferential padding. Within each of these domains, a number of circles of specified area was randomly placed. Area averages of $k^*$ and its increments in a circle were taken if at least 75 % of the circle area was in the domain and the circle included a number of pyranometers specified to maintain a constant sensor density. If there was an excess of pyranometers within a randomly-chosen circle, a subset was randomly selected to maintain the specified density. This sampling method was adopted because of the irregular distribution of sensors in the two campaigns. Three representative circles are illustrated for the Jülich campaign in Fig. 7, using circle radii of 1.25 km and a fixed sensor density of 2 km$^{-2}$.

The timeseries of $k^*$ and $\Delta k^*_\tau$ were then spatially averaged over the sensors within each circle for an ensemble of 500 different circles within each of ten logarithmically spaced area bins, using a constant sensor density of 2 km$^{-2}$. Samples were drawn from both field campaigns, with the Melpitz campaign only allowing the use of circles up to about 4.5 km$^2$, and the Jülich campaign covering the entire spatial range. Using the median standard deviations of $k^*$ and $\Delta k^*_\tau$ over all sensors as normalizing factors, we define the normalized variability of area-averaged quantities as the relative standard deviations

$$\hat{\sigma}^{k^*} = \frac{\sigma^{k^*}_{area}}{\sigma^{k^*}_0} \tag{8}$$

and

$$\hat{\sigma}^{\Delta k^*_\tau} = \frac{\sigma^{\Delta k^*_\tau}_{area}}{\sigma^{\Delta k^*_\tau}_0}. \tag{9}$$

Conditioned on the previously described sky types and time lags, summary statistics (cf. Table 1) of these relative variabilities are presented as a function of averaging area in Fig. 8. As circles with areas below 0.5 km$^2$ contain only single sensors, no averaging occurs in these circles and the median relative variability for this area is one. Although the standard deviation is not a good measure for extreme fluctuations (due to the non-Gaussian character of $k^*$ increment statistics), it offers a convenient way of characterizing "typical excursions from the mean" and has previously been used in similar contexts, for example by Hoff and Perez (2010, 2012). The standard deviation is also convenient to use because irrespective of the distribution of the data the relative variability will change as $n^{-0.5}$ if the sensors are uncorrelated, where $n$ is the number of sensors in the circle. The curve for the relative variability of uncorrelated sensors is included in Fig. 8.

Variability in averaged clearsky index decreases much more slowly with averaging area than does variability in increments. The decrease of variability with averaging area is also more rapid for shorter increment times than longer increments, and





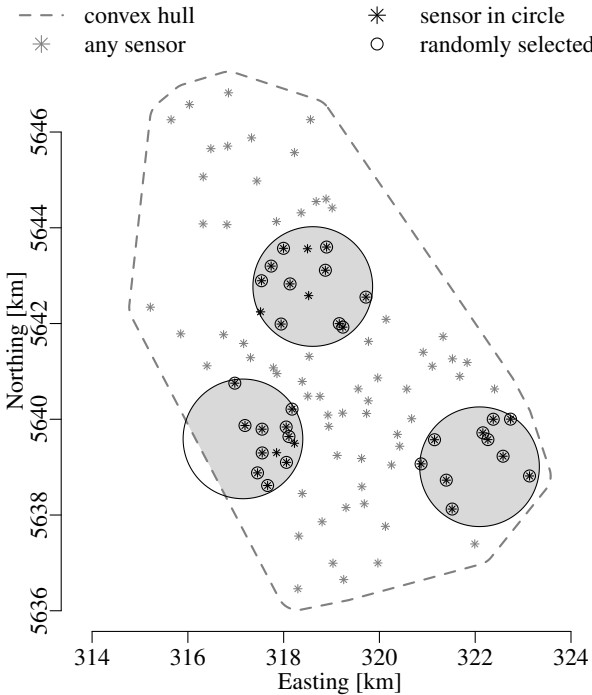

**Figure 7.** Three representative realizations of randomly selected circles falling within the study domain, using circle radii of 1.25 km, and a fixed sensor density of 2 km$^{-2}$. The coordinates (UTM projection, grid zone 32U) of all pyranometers of the Jülich campaign are shown, along with their corresponding convex hull, using a circumferential padding amounting to the mean minimum distance between sensors (about 440 m). Valid circles used to compute area-averaged quantities must overlap the domain by at least 75 % and include a specified number of pyranometers. If there is an excess of pyranometers within a randomly-chosen circle, random subset selection will ensure the adherence to a constant sensor density.

less rapid for overcast conditions than other sky types. For both $\tau = 1$ s and $\tau = 10$ s, the decrease of $\hat{\sigma}^{\Delta k^*_\tau}$ closely follows the $n^{-0.5}$ curve corresponding to uncorrelated sensors. This is also generally true for clear and mixed conditions, but not for $\tau = 60$ s under overcast conditions. This behavior is in agreement with the correlation structures presented in Fig. 4 and 6. As the standard deviation of the sum $x + y$ of two random variables with correlation coefficient $\rho_{xy}$ is

$$\sigma_{x+y} = \sqrt{\sigma_x{}^2 + \sigma_y{}^2 + 2\rho_{xy}\sigma_x\sigma_y}, \tag{10}$$

the standard deviation of the sum (or mean) of n positively correlated variables is always larger than that of n uncorrelated variables. For quantities with relatively long decorrelation distances, e.g. $k^*$ under overcast conditions (cf. panels A1 and B1 in Fig. 4), the effect of area averaging on fluctuations is reduced accordingly. The shorter the decorrelation distance becomes, the closer $\hat{\sigma}$ follows $n^{-0.5}$. Except for the aforementioned 60 s increments under overcast conditions, the decorrelation scales of increments are small compared to the radii of the circles used to compute the area averages. For example, averages for circles of diameter 1 km ($A \simeq 0.8$ km$^2$) and larger will be influenced when the decorrelation scale is about 1 km or smaller.





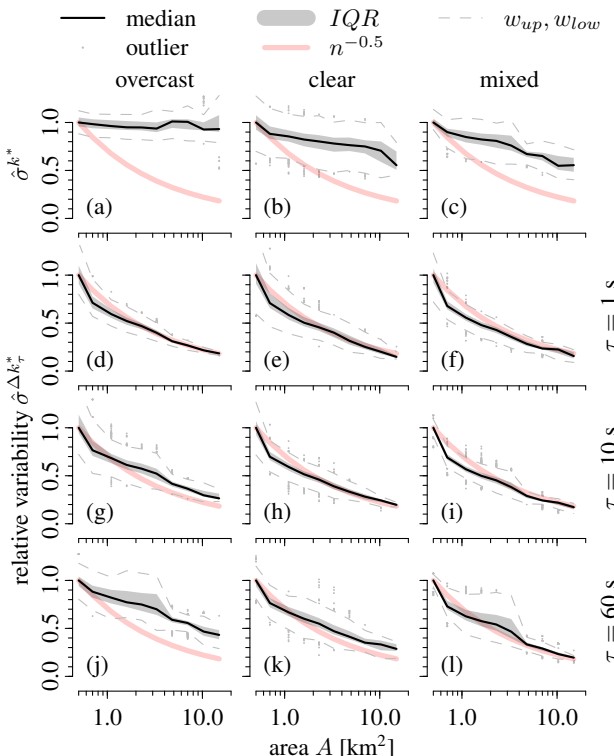

**Figure 8.** Normalized standard deviations $\hat{\sigma}^{k^*}$ and $\hat{\sigma}^{\Delta k^*_\tau}$ in clearsky index $k^*$ (panels a – c) and its increments $\Delta k^*_\tau$ (panels d – l) as functions of averaging area $A$ for different short-term time lags $\tau$, and conditioned on different sky types. The summary statistics (cf. Table 1) are computed for ten discrete areas that are logarithmically increasing in size, and contrasted with the uncorrelated decrease of variability following $n^{-0.5}$. The random circle method is used to sample these areas, using 500 different circles for each area bin, and a constant sensor density of 2 km$^{-2}$. Samples are drawn from both field campaigns, with the Melpitz campaign only allowing for circles up to about 4.5 km$^2$ of area, and the Jülich campaign covering the entire spatial range.

## 6 Discussion

When considering the implications of clearsky index variability for PV power, the short-term sky type classifications used throughout this study can be linked to distinct PV power fluctuation levels as in Perpiñán et al. (2013): overcast, clear, and mixed conditions correspond respectively to low, intermediate, and high PV fluctuation levels. While overcast conditions may necessitate other power sources to substitute the momentary lack of PV generation, clear and mixed conditions can negatively impact the electrical grid. Under clear conditions, high PV power feedback (i.e. reverse power flow from the distribution grid to the transmission grid) can occur and needs to be managed in areas of high PV penetration (Wirth et al., 2014), whereas mixed conditions may endanger the grid's reliability due to the frequent power ramping of PV systems (van Haaren et al., 2014). In





the context of sub-minute variability, we illustrate the event risk of fluctuations by means of increment PDFs in Fig. 5, which quantifies how mixed conditions feature flatter tails and higher probabilities of strong fluctuations than overcast and clear skies.

The relationship between measured irradiance and PV power often includes a smoothing effect, because the area of a pyranometer is very small compared to that covered by a PV system's panels. In general, the larger the spatial footprint of a PV plant, the more pronounced the smoothing effect will be, leading to a substantial difference between the variability characteristics of single utility scale PV plants covering relatively small areas, and fleets of distributed systems spanning relatively large areas (Hoff and Perez, 2010). The spatial scales covered by our datasets are representative of both distributed generation and utility scale power systems, and the spatial autocorrelation structures of $k^*$ and $\Delta k^*$ fields presented in Fig. 4 and 6, as well as the area averaged results of Fig. 8 quantify the smoothing effects. For a scenario of low-penetration distributed generation (two systems per km$^2$), the area-averaged variability in $k^*$ accordingly remained under the influence of positively correlated sensors (especially under overcast conditions). In contrast, variability in area-averaged sub-minute $\Delta k^*_\tau$ was shown to vary as one over the square root of the number of sensors as this quantity is only weakly correlated for the sampled area sizes in Fig. 8.

Hoff and Perez (2012) have used hourly satellite-derived irradiance data from three different locations in the United States to analyze two-point correlations of $k^*$ increments as a function of pair distance, with samples being separated between 10 and 300 km. Taking relative cloud speed and increment time lag ($1\,\mathrm{h} \leq \tau \leq 4\,\mathrm{h}$) into account, they proposed a model for $\rho_{ij}^{\Delta k^*_\tau}$ (cf. Eq. 7) that implies a linear relationship between distance and time lag for fixed values of the correlation coefficient. Hoff and Perez (2012) provide evidence for this relationship by presenting a linear scaling of station pair distances and time lags for fixed correlation coefficients, based on satellite-derived data (their Fig. 7). Perez et al. (2012) show similar results for decorrelation distances below 10 km and time lags below 15 minutes, based on virtual pyranometer networks (i.e. single sensor measurements shifted in time) with temporal resolutions of the single-point measurements being as low as 20 s (their Fig. 5). Our findings indicate that this linear scaling of distance and time lag may not necessarily hold for observed multi-point samples of $k^*$ fields at very high spatio-temporal resolutions. Along with the decorrelation distances $d_0$, Table 2 also quotes effective decorrelation length scales $d'_0 = \hat{d}_0 \cdot \tau$ for each time lag. These are derived using scaled decorrelation distances $\hat{d}_0$ obtained from the generalized analysis of correlation coefficients as functions of distance over time lag (Fig. 6 j – l). While the values of $d_0$ agree reasonably well with $d'_0$ for $\tau = 10\,\mathrm{s}$ and $\tau = 60\,\mathrm{s}$, the results differ substantially for $\tau = 1\,\mathrm{s}$. When analyzing correlation coefficients as functions of distance over time lag, as in Fig. 6 (j) – (l), the curves associated with the different sub-minute $\tau$ do not collapse on top of each other when plotted separately (not shown). Instead, high-$\tau$ data yield systematically lower correlation coefficients than low-$\tau$ data, and scaled decorrelation distances $\hat{d}_0$ are sorted in descending order along the 1 s, 10 s, and 60 s time lags. This sequencing has also been observed, but not discussed, by Hinkelman (2013) (her Fig. 9) for $10\,\mathrm{s} < \tau < 600\,\mathrm{s}$, which again suggests that the linear relationship between distance and time lag clearly shown by Hoff and Perez (2012) to govern large scales may not be applicable on very small scales.





## 7 Conclusions

With the continual global increase of PV power systems and the inherent weather-induced volatility of their power output, characterizing the underlying irradiance variability in both space and time is important for the planning and reliable operation of future power grids. In the present study, we analyzed spatio-temporal field characteristics of clearsky index and sub-minute $k^*$ increment variability during the HD(CP)$^2$ Observational Prototype Experiment HOPE for distances between tens of meters and about ten kilometers. The use of up to 99 synchronized silicon photodiode pyranometers operating at a temporal resolution of 1 Hz allowed characterization of variability in the lower ranges of the relevant space and time scales with unprecedentedly fine resolution.

By means of a simple classification scheme based on clearsky index statistics, we identified overcast, clear and mixed sky conditions, and subsequently analyzed sub-minute $k^*$ increments conditioned on these sky types. Mixed sky conditions, characterized by relatively high spread and intermediate averages of clearsky index, were shown to feature sub-minute increment PDFs with flatter tails and higher probabilities of strong fluctuations relative to overcast and clear skies. Of the three cloud types, mixed conditions are the most potentially problematic in terms of short-term PV power fluctuations. Compared to increment statistics computed without conditioning by sky type, the probability of relatively strong $k^*$ increments of $\pm 0.5$ was approximately twice as high under mixed conditions.

The corresponding spatial autocorrelation structures of $\Delta k_\tau^*$ revealed very low correlation coefficients for $\tau = 1$ s, even for short distances. As well, decorrelation distances under mixed and clear conditions were considerably shorter (e.g. 0.20 km and 0.23 km, respectively, for $\tau = 10$ s) than those under overcast skies (2.17 km for $\tau = 10$ s). As a proxy for the smoothing effects of distributed PV, spatial averaging was shown to effectively mitigate relative variability in $k^*$ increments with increasing areas. While, for example, averaging areas required to reduce $\hat{\sigma}^{\Delta k_\tau^*}$ in half were nearly the same for $\tau = 1$ s (1.9 km$^2$, 1.7 km$^2$, and 1.5 km$^2$ respectively for overcast, clear, and mixed skies), they differed more between overcast conditions on the one hand, and clear or mixed skies on the other hand, with increasing $\tau$ (for $\tau = 10$ s: 3.6 km$^2$, 1.8 km$^2$, and 1.6 km$^2$; for $\tau = 60$ s: 9.1 km$^2$, 2.9 km$^2$, and 2.8 km$^2$ respectively for overcast, clear, and mixed skies). Spatial averaging on the scales under consideration was less effective at mitigating variability in $k^*$.

These initial characterizations of PV-related clearsky index variability during HOPE can be extended to consider other issues of relevance to solar PV power generation. For example, a more refined analysis of the two-point spatial autocorrelation structures of $k^*$ increments could be carried out using actual cloud speed information, and the linear scaling of distance and time lag discussed in section 6 could be investigated in detail. Combining these data with coarser resolution ones, for example satellite-derived, could thereby appreciably extend the spatial domain of the analysis. An evaluation of short-term variability reduction due to temporal averaging could also be a direction of future study, taking advantage of the unusually high temporal resolution of the original data acquisition unit (10 Hz) to critically assess the common practice of using longer temporal averages when measuring irradiance (Ohmura et al., 1998).





*Acknowledgements.* We thank Andreas Macke at the Leibniz Institute for Tropospheric Research TROPOS (Leipzig, Germany) for sharing the pyranometer network datasets of the HD(CP)[2] Observational Prototype Experiment HOPE, and acknowledge helpful comments from Annette Hammer. The time and effort invested in developing and maintaining R (version 3.2.2) by the R Core Team (2015) and the active community of package authors is also gratefully appreciated. This research was partially funded by the Lower Saxony research network
5   'Smart Nord', which acknowledges the support of the Lower Saxony Ministry of Science and Culture through the 'Niedersächsisches Vorab' grant program (grant ZN 2764 / ZN 2896). It was also partly funded by the 'Performance Plus' research project through the European Union's Seventh Framework Program for research, technological development and demonstration (grant agreement no. 308991). We also acknowledge funding support from the Natural Sciences and Engineering Research Council of Canada.





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
