# Peer review of "Local short-term variability in solar irradiance"

_Atmospheric Chemistry and Physics, 2016_

## Referee Comment (RC1) · O. Perpiñán (Referee) · 6 Mar 2016

In my opinion, this paper is of excellent quality. Data and methods are clearly exposed, including several engaging ideas. Data analysis is exhaustive, and the results are shown with a collection of superb figures. I think this paper will be an important contribution to the solar radiation variability subject, both for its methods, results, and discussion, and for the quality of the paper itself.

However, there are some issues that could be clarified or improved:

- The authors underline the time resolution and the spatial density of the data. However, the duration of the data is very short (4 months and 1 month, respectively) compared with papers previously published. Moreover, the time series only

cover a certain period of the year (spring and autumn). These characteristics may limit the conclusions inferred from the time series.

- Measurements correspond to global irradiation on the horizontal plane (GHI). However, PV plants produce power with solar irradiance on a inclined plane. It must be noted that, at least on a daily basis, the variability of the effective irradiation incident on inclined planes has been reported to be higher than the variability of irradiation on the horizontal plane:

    – Suri, M., Huld, T., Dunlop, E.D., Albuisson, M., Lefevre, M., Wald, L., 2007. Uncertainties in photovoltaic electricity yield prediction from fluctuation of solar radiation. In: 22nd European Photovoltaic Solar Energy Conference.
    – Perpinan, O., 2009. Statistical analysis of the performance and simulation of a two-axis tracking PV system. Solar Energy 83 (11), 2074–2085.

- In order to remove trends in GHI variability, the authors compute the clear sky index from the GHI measurements. The problem with this index is that the subsequent results are model dependent. In fact, there is not a unique clear sky index because there are several clear sky models to choose. Moreover, most of the models require the use of aerosol measurements or estimations, or assumptions regarding the atmospheric conditions. Therefore, the clear sky model imposes additional uncertainties that were not present in the original data.

- The paper includes a good bibliographic review in the introduction section. However, afterwards the results of the paper are not related to the reviewed papers, and the analysis does not put the results in the context of that review. For example, the figures 4, 5, 6, and 8 show similarities with the figures published in some papers included in the bibliographic review, but there is almost no comments about it.

- Equation 6 uses a simple increment to compute the fluctuations of k*. This approach could be improved as discussed (for example) in Gallego, Cristóbal, Alexandre Costa, Álvaro Cuerva, Lars Landberg, Beatrice Greaves, and Jonathan Collins. 2013. "A Wavelet-Based Approach for Large Wind Power Ramp Characterisation." Wind Energy 16 (2): 257–78.

- The paper will be greatly improved if the authors could publish both the measurements data and the R code, following the recommendations on Reproducible Research: "When publishing computational results, including statistical analyses and simulation, provide links to the source-code (or script) version and the data used to generate the results to the extent that hosting space permits." The Yale Law School Roundtable on Data and Code Sharing. 2010. "Reproducible Research." Computing in Science & Engineering 12 (5). Los Alamitos, CA, USA: IEEE Computer Society: 8–13.

---

## Referee Comment (RC2) · Anonymous Referee #2 · 18 Mar 2016

I agree with the comments listed by Reviewer 1. In addition:

• This paper examines clear sky irradiance at 1 Hz. Given the response time of solar photovoltaics and solar photothermal, is this frequency necessary? I suspect an argument for this could go either way; please include a discussion in the discussion/conclusions.

• The finding that of a very low spatial autocorrelation at tau=1 is not surprisingly, given the physical nature of clouds. Indeed, previous papers cited by the author suggest that the spatial autocorrleation should be low even at 5 or 15 minute time steps. Why is this important to have discovered, and what does it show beyond what we already know?

• A similar question with the decorrelation. . . Wouldn't it make sense that the decorrelation distances would be a function of clouds? In an area of almost entirely one climatic zone, cloud conditions in one location will (almost by definition) be correlated with other locations, and thus the distance needed to obtain decorrelation will increase. It is not clear this is novel; at a minimum, the authors should describe why this finding grees entirely with what one would expect from real time data.

---

## Author Comment (AC1) · 20 Apr 2016

Thank you very much for reviewing our manuscript. We appreciate the positive feedback and feel that the detailed comments have helped us to improve the quality of the paper.

Below, each comment is quoted in italics and followed by its respective author response. A corresponding revised version of the manuscript is attached to this response as a supplement. It has been prepared by means of latexdiff and highlights all differences between the original and revised versions of the paper. All page and line numbers quoted below refer to this supplement file.

*The authors underline the time resolution and the spatial density of the*

[Figure]

*data. However, the duration of the data is very short (4 months and 1 month, respectively) compared with papers previously published. Moreover, the time series only cover a certain period of the year (spring and autumn). These characteristics may limit the conclusions inferred from the time series.*

We have included corresponding caveats in sections 2.1 and 7 (Measurement campaigns and Conclusions) on page 3 (lines 30–31), and page 20 (lines 10–11), respectively.

*Measurements correspond to global irradiation on the horizontal plane (GHI). However, PV plants produce power with solar irradiance on a inclined plane. It must be noted that, at least on a daily basis, the variability of the effective irradiation incident on inclined planes has been reported to be higher than the variability of irradiation on the horizontal plane:*

- *Suri, M., Huld, T., Dunlop, E.D., Albuisson, M., Lefevre, M., Wald, L., 2007. Uncertainties in photovoltaic electricity yield prediction from fluctuation of solar radiation. In: 22nd European Photovoltaic Solar Energy Conference.*
- *Perpinan, O., 2009. Statistical analysis of the performance and simulation of a two-axis tracking PV system. Solar Energy 83 (11), 2074–2085.*

We have added both references and included a corresponding statement in section 6 (Discussion) on page 18 (lines 27–29).

*In order to remove trends in GHI variability, the authors compute the clear sky index from the GHI measurements. The problem with this index is that the subsequent results are model dependent. In fact, there*

*is not a unique clear sky index because there are several clear sky models to choose. Moreover, most of the models require the use of aerosol measurements or estimations, or assumptions regarding the atmospheric conditions. Therefore, the clear sky model imposes additional uncertainties that were not present in the original data.*

We have extended the corresponding paragraph in section 2.2 (Clearsky index) on page 5 (lines 5–8) accordingly.

*The paper includes a good bibliographic review in the introduction section. However, afterwards the results of the paper are not related to the reviewed papers, and the analysis does not put the results in the context of that review. For example, the figures 4, 5, 6, and 8 show similarities with the figures published in some papers included in the bibliographic review, but there is almost no comments about it.*

We have added comparative comments and cross references to the anaylses of our figures 5, 6, and 8 in sections 4 and 5 on pages 11 and 12 (lines 10–11 and 1–2), page 14 (lines 4–12), and page 18 (lines 9–16). As for figure 4, we don't see immediate similarities with the cited literature. Our figure characterizes the spatial autocorrelation structures of short clearsky index time series for different values of their mean and spread, while the reviewed papers focus on the direct analysis of $k^*$-increment properties.

*Equation 6 uses a simple increment to compute the fluctuations of k\*. This approach could be improved as discussed (for example) in Gallego, Cristóbal, Alexandre Costa, Álvaro Cuerva, Lars Landberg, Beatrice Greaves, and Jonathan Collins. 2013. "A Wavelet-Based Approach for Large Wind Power Ramp Characterisation." Wind Energy 16 (2): 257–78.*

We shall consider this suggestion for future analyses and have consequently mentioned it as an outlook in section 7 (Conclusions) on page 21 (lines 6–7).

> *The paper will be greatly improved if the authors could publish both the measurements data and the R code, following the recommendations on Reproducible Research: "When publishing computational results, including statistical analyses and simulation, provide links to the source-code (or script) version and the data used to generate the results to the extent that hosting space permits." The Yale Law School Roundtable on Data and Code Sharing. 2010. "Reproducible Research." Computing in Science & Engineering 12 (5). Los Alamitos, CA, USA: IEEE Computer Society: 8–13.*

We understand the idea behind this comment and would of course like to contribute to the reproducibility of our results. However, we feel that the availability of the source code is less important for this purpose than the availability of the irradiance dataset. The procedures of our analyses are described in great detail in the text, and the methods are comparatively straight forward to apply (basically, it's just a great many additions, subtractions, multiplications and divisions; combined with different conditionings). Thus, we consider the current work reproducible without the original source code, but will consider preparing the code of future analyses in a more open manner. As for the data, we have added an unnumbered section "Data availability" before the acknowledgements on page 21 (lines 8–10), containing a reference to the appropriate project website, from which the HD(CP)$^2$ data portal is by now accessible.

Please also note the supplement to this comment:
http://www.atmos-chem-phys-discuss.net/acp-2016-2/acp-2016-2-AC1-supplement.pdf

---

## Author Comment (AC2) · 20 Apr 2016

Thank you very much for reviewing our manuscript. We appreciate the feedback and feel that the comments have helped us to improve the quality of the paper.

Below, each comment is quoted in italics and followed by its respective author response. A corresponding revised version of the manuscript is attached to this response as a supplement. It has been prepared by means of latexdiff and highlights all differences between the original and revised versions of the paper. All page and line numbers quoted below refer to this supplement file.

*This paper examines clear sky irradiance at 1 Hz. Given the response time of solar photovoltaics and solar photothermal, is this frequency neces-*

[Figure]

*sary? I suspect an argument for this could go either way; please include a discussion in the discussion/conclusions.*

On the one hand, typical solar thermal energy (STE) systems are intentionally designed with storage capacities, and 1 Hz irradiance data is probably not necessary for the majoroity of applications involving STE.

On the other hand, the response of a single photovoltaic cell to changes in its illumination is orders of magnitude faster than 1 Hz and thus virtually instantaneous within the scope of our analyses. Of course, when aggregating many cells in a PV module, and then connecting a large number of modules in a PV system, spatial smoothing increases the system's response time. For very many inter-connected PV systems in a very large area, e.g. all of Europe, the necessary temporal resolution of data is hence appreciably reduced (the European Energy Exchange, for example, uses 15 minute time steps for electricity trading). However, Marcos et al. (2011b) find rare power fluctuations of e.g. up to $\pm$ 50 % from one second to the next (and changes of more than 90 % for a time lag of 20 s) in 1 Hz power data from a single, relatively small, 48 kWp PV plant (typical rooftop systems are smaller yet and experience less smoothing). Yordanov et al. (2013b) even argue that the optimal temporal resolution of single point measurements should be around 10 Hz, in order not to miss extremely short but relatively high magnitude changes. Thus, a temporal resolution of 1 Hz may not be necessary for large-scale analyses, but it is key to characterize local short-term variability in solar irradiance on the spatio-temporal scales that we investigated.

We have extended the corresponding paragraphs in section 6 (Discussion) on page 18 (lines 30–32) and page 19 (lines 1–11) in line with the above answer.

> *The finding that of a very low spatial autocorrelation at tau=1 is not surprisingly, given the physical nature of clouds. Indeed, previous papers cited by the author suggest that the spatial autocorrleation should be low even at*

*5 or 15 minute time steps. Why is this important to have discovered, and what does it show beyond what we already know?*

We agree that very low spatial autocorrelation coeffcients of clearsky index increments $\rho_{ij}^{\Delta k_\tau^*}$ are indeed no surprise for $\tau = 1$ s. However, we rather see the reason for this in the smallest inter-sensor separation bin (ranging from 28 m to 51 m) being much greater than typical cloud speeds (between 2 ms$^{-1}$ and 10 ms$^{-1}$). The cloud-induced shadows would frequently have to cover the shortest of the sensor pair distances within a second, in order to yield high values of $\rho_{ij}^{\Delta k_\tau^*}$ at short distances. For a robust characterization of the decrease of $\rho_{ij}^{\Delta k_\tau^*}$ from 1 to 0, the pyranometer network would have to be reconfigured to feature much shorter inter-sensor distances. We have inserted a short paragraph along these lines at the appropriate place in section 7 (Conclusions) on page 20 (lines 21–25) to make ourselves more clear.

We disagree, however, that previously published papers would generally suggest $\rho_{ij}^{\Delta k_\tau^*}$ to yet be low for time lags of 300 s or even 900 s on the small spatial scales considered by our analysis. For example, the virtual networks studied by Perez et. al. (2012) on average suggest positive station pair correlation coefficients of $1.0$ at distances of 100 m for these time lags, and correlation coefficients of $\sim 0.6$ ($\sim 0.75$) for distances of 1000 m for 300 s (900 s) increments (see their Fig. 4). Likewise, the satellite-derived model by Hoff and Perez (2012) used for comparison in our Fig. 6 predicts correlation coefficients for e.g. 1000 m distances to range from 0.375 through 0.75 for time lags of 300 s, and from 0.643 through 0.9 for 900 s (using effective wind speeds between 2 ms$^{-1}$ and 10 ms$^{-1}$). In good agreement with the above, Hinkelman (2013) shows 300 s increments to be associated with $\rho_{ij}^{\Delta k_\tau^*} \simeq 1$ for distances of about 100 m and $\rho_{ij}^{\Delta k_\tau^*} \simeq 0.4$ for distances of about 1000 m, based on data from a small pyranometer network during 13 days of broken clouds.

*A similar question with the decorrelation... Wouldn't it make sense that*

*the decorrelation distances would be a function of clouds? In an area of almost entirely one climatic zone, cloud conditions in one location will (almost by definition) be correlated with other locations, and thus the distance needed to obtain decorrelation will increase. It is not clear this is novel; at a minimum, the authors should describe why this finding grees entirely with what one would expect from real time data.*

We agree that decorrelation distances of $\Delta k_\tau^*$ greatly depend on cloud field properties. In previous analyses for example, Hoff and Perez (2010 and 2012) argue that (de)correlation of $k^*$ increments should mainly be a function of (1) distance, (2) time scale, and (3) effective cloud speed, while Perpinan et al. (2013) show spatial autocorrelation structures of PV power fluctuations to depend on (1) distance, (2) time scale, and (3) one of three daily 'fluctuation categories' (which are comparable to our sky types). The distinction between sky types is thus not what is novel about our study. What is novel, however, is the detailed characterization of $\rho_{ij}^{\Delta k_\tau^*}$ at very high spatio-temporal resolutions, including – but not limited to – decorrelation distances under different sky conditions. We have removed the single sentence emphasizing the varying decorrelation distances under different sky conditions from section 7 (Conclusions) on page 20 (lines 20–21) in order to avoid potential misunderstandings in terms of what's novel.

Please also note the supplement to this comment:
http://www.atmos-chem-phys-discuss.net/acp-2016-2/acp-2016-2-AC2-supplement.pdf

[Figure]

**Supplement:**

[revised manuscript text omitted]